# Physical Fitness Characteristics of High-level Youth Football Players: Influence of Playing Position

**DOI:** 10.3390/sports7020046

**Published:** 2019-02-16

**Authors:** David Bujnovky, Tomas Maly, Kevin R. Ford, Dai Sugimoto, Egon Kunzmann, Mikulas Hank, Frantisek Zahalka

**Affiliations:** 1Faculty of Physical Education and Sport, Charles University, 162 52 Prague, Czech Republic; bujnovsky@ftvs.cuni.cz (D.B.); egonkunzmann@gmail.com (E.K.); hank@ftvs.cuni.cz (M.H.); zahalka@ftvs.cuni.cz (F.Z.); 2Department of Physical Therapy, High Point University, High Point, NC 27268, USA; kford@highpoint.edu; 3The Micheli Center for Sports Injury Prevention, Waltham, MA 02453, USA; dai.sugimoto@childrens.harvard.edu; 4Department of Orthopaedic Surgery, Harvard Medical School, Boston, MA 02115, USA; 5Division of Sports Medicine, Department of Orthpaedics, Boston Children’s Hospital, Boston, MA 02115, USA

**Keywords:** performance demands, elite sport, GPS, match analysis, agility, endurance, testing and diagnostic

## Abstract

The aim of this study was to determine whether the speed, agility, aerobic and anaerobic capacities of football players varied by playing positions. Elite youth football players (*n* = 123, age = 15.7 ± 0.5 years) who played in six different positions, as goalkeepers (GK), full backs (FB), central defenders (CD), wide midfielders (WM), central midfielders (CM), and attackers (AT), were assessed. Multivariate analysis of variances was used to compare the following variables: Linear running sprint for 5 m (S5) and 10 m (S10), flying sprint for 20 m (F20), agility 505 test with turn on the dominant (A505D) and non-dominant leg (A505N), agility K-test, Yo-Yo intermittent recovery (YYIR1) test and repeat sprint ability (RSA) test. The results showed significant influence of playing positions on linear-running sprint performance (F_1,123_ = 6.19, *p* < 0.01, η_p_^2^ = 0.23). Midfielders reached significantly higher performance levels (CM = 2.44 ± 0.08 s, WM = 2.47 ± 0.13 s) in the A505N test compared to GK (2.61 ± 0.23 s). Outfield players had significantly higher performance in both YYIR1 and RSA tests compared to GK (*p* < 0.01). The results of this study may provide insightful strategies for coaches and clinical practitioners for developing position-specific conditioning programs.

## 1. Introduction

In recent years, football success has been shown to be highly dependent on various physical, technical, tactical, and psychological factors [1,2,3,4,5,6,7,8]. Bangsbo [9] emphasized that for successful competitiveness, the development of speed, agility, strength with combination of aerobic and anaerobic (even maximal) abilities is important for successful, competitive football careers.

It is crucial that individual player position requirements be considered during football practice as pertains to completing tactical tasks, such as tandem defending, attacking runs behind the defensive line, and high pressing. Players require specific skills and superior physical conditioning in order to effectively execute these tasks. Differences in the evaluation of player parameters have been shown to correlate with playing positions, as observed in many studies with respect to different parameters: Differences in total distance covered [5,10,11], differences in sprint distance [10,12,13], isokinetic strength [14,15,16], morphological and body composition [17,18], power assessment [19,20] and VO2max uptake [21,22].

Full backs (FB) have higher aerobic endurance performance than both central defenders (CD; up to 17%) and attackers (AT) (14%) [23]. Additionally, midfielder players have better performance than CDs (13%) [23]. Krepsi et al. [24] provided a running intervention to football players and found that midfielders demonstrated the most changes in sprinting ability compared to other field position players. The greater adaptation could be explained by the fact that midfielders are engaged in significantly less walking and low-intensity runs, but spend the most time running and sprinting [25]. Midfielders also have higher levels of maximal oxygen uptake (VO2max) and run greater distances compared to other positions [26].

Recent investigations [11,27,28] and an earlier study conducted by Matković et al. [29] emphasized that age, morphology, and physical fitness were influential parameters of football performance in elite-level football players, but also confirmed that playing position decisively dictated absolute performance loads and the intensity of fast movements during matches. Although these data are valuable, evidence at the youth level is considerably lacking, especially at a competitive level. Investigating how physical fitness parameters differ based on various playing positions in high-level, young football players may have a profound influence on their daily training and future performance. Sarmento et al. [30] reported that collecting and measuring a large volume of data (e.g., positional, physiological, psychological, environmental conditions, etc.) in real time, and compressing it into a smaller set of variables, providing objective information for coaches that facilitates, to some extent, the prediction of performance outcomes, seems to be a useful path in this specific area.

Therefore, the aim of this study was to investigate the differences in the various physical fitness characteristics of high-level, youth football players.

## 2. Materials and Methods

### 2.1. Study Design

A cross-sectional study design was used in this study. The research was approved by the ethical committee of the Ethical Committee, Faculty of Physical Education and Sport, Charles University (Nr. 101/2018). Measurements were recorded in accordance with the ethical standards of the Declaration of Helsinki and ethical standards in sport and exercise science research [31]. Signed informed consent was collected before the various assessments were performed.

### 2.2. Subjects

The research participants consisted of 123 players of the top division of the Czech football league (*n* = 123, age = 15.7 ± 0.5 years, body height, BH = 178.5 ± 6.8 cm, body mass, BM = 68.2 ± 8.4 kg) who were clustered by their playing position: Goalkeeper (GK, *n* = 9), full back (FB, *n* = 25), central defender (CD, *n* = 15), wide midfielder (WM, *n* = 27), central midfielder (CM, *n* = 25), and attacker (AT, *n* = 22).

### 2.3. Procedures

#### 2.3.1. Anthropometric Measurement

Anthropometric data were recorded before body composition assessment. BH was measured using a digital stadiometer (SECA 242, Seca, Hamburg, Germany) and BM was collected using a digital scale (SECA 769, Seca, Hamburg, Germany).

#### 2.3.2. Assessment of Physical Fitness Characteristics

##### Linear Sprint Running

Speed indicators were assessed using field motor tests of running speed. Performances in linear running sprints for 5 m (S5), 10 m (S10) and a flying sprint for 20 m (F20) after a 30-m run-up were measured using photocells (Brower Timing System, Brower, Draper, UT, USA). In the sprint speed test, the players ran a 10-m distance; however, 5-m performance was also measured. To assess maximal speed, we used a flying 20-m sprint after a 30-m run-up, which had also been used for senior professional players [32].

##### Agility

To evaluate agility, we used the 505 test [33], which includes both acceleration and deceleration phases of a run with a 180° turn on the dominant (A505D) or non-dominant leg (A505N). This test was used for a single change of direction speed. For assessment of repeated changes of direction speed, we used the K-test [34], in which a player runs through spaces between cones arranged in a “K” shape at maximum speed (Figure 1). The cones were 35 cm high, each with a contact switch on top with a diameter of 7 cm. The player stood by the middle cone (no. 1), and after starting on his own (pushing the switch), he ran to cone no. 2, where again he tapped the switch with his hand and then ran back to the initial cone (no. 1). High scores were reported for session reliability in young athletes for the agility K-test [35].

##### Repeated Sprint Ability (RSA)

Physiological responses to intermittent exercise were assessed with the RSA test [36]. The test depended primarily on the ability to recover from the work period with a change of direction speed ability. Performance depended on the duration of repetitions, the rest duration, and the number of repetitions performed in the work session. The goal of the player was to complete repeated sprints (8 times) with a change of direction to the right and left sides (Figure 2). At the end of each run, the player returned (recovery) and ran again from the starting position. Each run started after 30 s interval. The total time for the procedure was 4 min (8 × 30 s). Performance was evaluated as an average of all 8 repeated runs.

##### Aerobic Capacity

To examine maximum aerobic capacity, the Yo-Yo intermittent recovery level 1 (YYIR1) test was used. This field test is focused on assessment of running distance and the aerobic capacity of football players [37]. The YYIR1 test consists of repeated lengths of 2 × 20 m, with players running back and forth in response to audio signals. Each player was tested in his own area, which was 2 m wide and 20 m long, with 5 m of active recovery. This area was marked by cones. The total time of testing did not exceed 20 min. After completing 2 × 20 m, players had 10 s of active recovery which consisted of 2 × 5 m walking or jogging. If the player failed to reach the line twice in the given time interval, the test was stopped and the total distance is recorded. The YYIR1 test consisted of 4 runs (2 × 20 m) at 10–13 km·h^−1^ (0–160 m) followed by 7 runs at 13.5–14 km·h^−1^ (160–440 m); after which the speed was increased by 0.5 km·h^−1^ for each of the 8 runs (meaning 760, 1080, 1400, 1720 m, etc.) up to the maximum capability. We used the total running distance for test performance assessment.

##### Statistical Analysis

For statistical processing of data, we first used descriptive statistics. Measurements and measures of variability were expressed as arithmetic means and standard deviations, respectively. Parametric procedures were chosen after data distribution verification of normality using the Shapiro–Wilk test. The assumptions for using a parametric test were satisfied, and differences in the observed dependent variables among the groups were assessed using multivariate analysis of variance (MANOVA). We used multiple comparisons of means (Bonferroni’s post hoc test) to compare differences in particular parameters among the groups.

We rejected the null hypothesis at the *p* ≤ 0.05 level. The “partial eta square” coefficient (η_p_^2^), which explains the proportion of variance of the factor, was used to assess effect size as follows: η_p_^2^ = 0.02 is considered a small effect, η_p_^2^ = 0.13 is considered a medium effect and η_p_^2^ = 0.26 is considered a large effect [38]. Statistical analyses were carried out using IBM^®^ SPSS^®^ v21 (Statistical Package for Social Sciences, Inc., Chicago, IL, USA, 2012).

## 3. Results

MANOVA revealed significant differences in observed parameters with respect to the playing position of football players (Wilks’ Λ = 0.18, F = 4.10, *p* < 0.01, η_p_^2^ = 0.29). We found the independent variable (playing position) to have significant effects on BH (F_1,123_ = 8.28, *p* < 0.01, η_p_^2^ = 0.26) and BM (F_1,123_ = 5.99, *p* < 0.01, η_p_^2^ = 0.20). Results also showed the significant influence of playing position on F20 performance (F_1,123_ = 6.19, p < 0.01, η_p_^2^ = 0.23). Bonferroni post hoc analysis revealed significantly better F20 performance in WM (2.34 ± 0.08 s) compared to GK (2.55 ± 0.11 s), CM (2.46 ± 0.11 s), and AT (2.44 ± 0.13 s). Furthermore, GK had significantly lower F20 performance compared to FB (2.38 ± 0.09 s) and CD (2.38 ± 0.09 s). Additionally, results showed a greater change of direction speed with turning on the non-dominant leg (A505N) in midfielders than GK. In terms of aerobic capacity (YYIR1) and repeated sprint ability, we found significantly better performance in all field position players (FB, CD, WM, CM, and AT) than GK (Table 1). Of the outfield players, FB had the best repeated sprint ability (5.46 ± 0.19 s) and CD (5.68 ± 0.19 s) and CM (5.68 ± 0.19 s) had the worst results.

## 4. Discussion

The purpose of this study was to investigate the differences in the various physical fitness characteristics of high-level, youth football players.

Our accelerated sprints (S5 and S10) findings were consistent with a recent study by Gilet et al. [39] (1.06 ± 0.1 s and 1.77 ± 0.1 s vs. 1.13 ± 0.07 s and 1.9 ± 0.1 s). Lockie et al. [40] support that linear speed is a crucial factor for football players for positional play and goal scoring. Elite players in their study showed significantly higher results than in our evaluation (S5 = 0.979 ± 0.046 s; S10 = 1.69 ± 0.059 s). However, the average age of the players analyzed in the study by Lockie et al. [40] was 21.2 ± 1.32 years (compared to the players in our study with an average age of 15.7 ± 0.5 years). The completion time for the 20 m sprint, and the 0–10 m and 10–20 m splits had a moderate to high correlation with the age of the athlete (*r* = −0.53, −0.40, and −0.57, respectively; *p* < 0.001). The older players scored more times than the younger players [41]. Pyne et al. [42] reported a completion time of 3.04 ± 0.08 s for the 20-m sprint from a stationary (crouched) start, which is approximately 0.6 s slower than that of our flying start 20-m sprint test. A strong correlation (*r* = 0.91) was also previously reported between both low-acceleration linear running sprints (S5, S10) in elite, youth football players [43].

In our study, CM had superior results for the A505D and A505N tests (1), but was not significantly different between playing positions. As midfielders usually operate in the middle of the field, they are required to constantly switch their direction of movements quickly. Precisely, midfielders are often responsible for both offensive and defensive tactics; thus, a quick transition is often required. To compare with other sports, Fernandez [44] conducted the 505 agility test on youth tennis players and obtained a result of 2.88 ± 0.17 s, which is worse performance than with the CM observed in our study (2.44 ± 0.08 s). Rapid change of direction speed in relatively small areas is important for tennis players as it is for football players and particularly so for CM. Significant but moderate correlations (*p* < 0.05) were found between all horizontal jump tests and the modified agility test or the 505 test, and between all vertical jump tests [45]. The agility test was administered and demonstrated a low correlation with the 20-m sprint test [46]. A study performed by Sheppard et al. [47] confirmed that agility correlates with trainable physical qualities such as strength, power, and technique and cognitive components such as visual-scanning techniques, visual-scanning speed, and anticipation.

During matches, players, irrespective of their playing positions, reach running speeds with velocities higher than 15 km·h^−1^ in up to 28% of a 10–12 km distance [48]. Regarding intensity, Bloomfield et al. [25] documented that AT performed the greatest amount of high-intensity running and CD are the most active in carrying out backward movements.

The ability and level of fast repeated sprint performance during matches are two of the most crucial determinants of sufficient physical preparation in players [49,50]. Physiological responses to intermittent exercise were noted in the RSA test. In our study, the best results were observed in FB (5.46 ± 0.19 s) and the worse scores were seen in CM and CD (5.68 ± 0.22 s and 5.68 ± 0.19 s, respectively) and GK (6.05 ± 0.18 s), which is not unusual considering the characteristics of the playing positions. The study by Impelizzeri et al. [51] tracked several changes (from short-term periods of 48 h to long-term/seasonal periods) from the point of the absolute reliability of the elite players in RSA (6 × 20 m sprint; 20 s rest between sprints). WM had the best result (6.83 ± 0.22 s) but CM had the worst result (7.01 ± 0.23 s). In our study, WM had the second best performance in RSA (5.54 ± 0.15 s).

The study by Deprez et al. [52] (*n* = 10) showed YYIR1 test results of 2404 ± 347 m. Mujika et al. [53] suggested that the YYIR1 test performance of approximately 2400 m should be the standard for elite male football players. The results obtained for the players in our study were about 2230 m and were similar for all outfield positions while GKs recorded significantly lower values (1815 m). Krustrup et al. [23] reported that the YYIR1 test strongly correlates with aerobic and anaerobic capacities of tested players and is closely linked to game performance. Interestingly, no significant differences were found in this study among field positions (FB, CD, WM, CM, and AT). However, the difference was noted between all of the field position players and GK. Svensson and Drust [54] pointed out that fitness tests in conjunction with physiological data should be used for monitoring changes in fitness and for guiding their training regime, but different scenarios are often encountered. Results of the study by Casajús et al. [55] concluded that regular and periodic performance evaluation of elite football players have proved to be rather difficult. Of the 21 members of the team, 18 took the first test, 17 took the second, 15 took the first and second tests, but only 11 went through all the tests in both phases. An excessive number of matches, frequent injuries, constant travelling, and the lack of a habit of undergoing physiological tests in laboratory conditions made this task difficult. The study by Di Salvo et al. [13] emphasized that CM covered the most distance during matches with increase in speed and covered distance even in the second half. These results agree with those from the earlier study by Bujnovsky et al. [14] which investigated the internal load (heart rate monitoring) in elite adult players. They reported that the highest demands of physical load at high intensity (over anaerobic threshold) were found in CM (21.8 ± 7.8%) and AT (17.8 ± 3.8%). On other hand, studies by Boone et al. [28] and Di Salvo et al. [13] reported the most covered distance by elite male football players in WM (11,410–11,500 m).

As the demands on football player performance increase with age, division of play or level of game, focused power and strength development grows in importance [56]. The recent study by Barrett et al. [2] with elite English Premier League players highlighted that FB had significantly higher ratings of perceived exertion, which can be considered an evaluation of internal load, than other players. In our study, FB had the best results in YYIR1 and RSA tests (2275.52 ± 205.34 m, 5.46 ± 0.19 s, respectively). The model and strategy of physical preparation must be as specific as possible. This means that, for instance, sprinting ability should not be trained by just sprinting, but the training should be conducted for the specific playing position in conditions as close to real match conditions as possible [13,56]. Measuring of physical performance could be useful in predicting later success in soccer [57]. In addition to the association between physical performance and successful soccer career development, the current study may serve a role of identifying an optimal window of different training modes based on positions. Furthermore, recent articles synthesized physical training and brain development in the youth population [58,59]. Another article suggested education and instruction of fundamental movements with progressive exercise formats through a variety of exercise modes and adequate rest in the pre-adolescent developmental stage help in enhancing athletic performance [60]. The current study data are likely feasible to construct an effective training tactics based on physical maturation/developmental stages and soccer positions.

## 5. Limitations

Despite the presented results, there are several limitations. The research was carried out on artificial grass (to standardize conditions for all players), without the stress factor of match play and also without a ball (non-specific field testing). In further research, it would be desirable to supplement the measurement with a ball and compare both general and specific results with a performance gained from a match or more specific tasks (e.g., small side games). Additionally, we used a group of high-level youth soccer players (AC Sparta Prague soccer academy). Thus, the findings of this study may not be generalizable to all soccer players, especially in the female population. Future studies are warranted to include soccer players with lower performance levels and female players. 

## 6. Conclusions

The results of this study showed significant differences in flying sprint performance between the tested groups of WM, CM, AT, and GK, with WM demonstrating significantly better ability than the other groups and GK showing significantly slower performance than CD and FB. Furthermore, midfielders performed better than GK in the A505N test and GK had significantly lower completion times than outfield players in the YYIR1 test. FB demonstrated the best RSA performance, which may be a result of common tactical play in the modern game of football, where they must complete a lot of repeated sprints to finally pass or provide a cross ball into penalty areas for facilitate attacking. Among the various position players, CM and CD demonstrated the worst RSA results.

Current results showed significant differences between the playing positions and the physical fitness conditions required for each position. For this reason, it is essential to work on the individual needs of players according to their positions during the football training process to help them achieve the required fitness levels necessary to perform efficiently on match days. The task of all coaches, including strength and conditioning coaches and athletics coaches, is to improve players’ condition and build successful teams. The results of the work of coaches can be beneficial to the sport in providing insight on how best to design individualized conditioning training.

In addition, it turns out that there is a need to review "traditional" field tests that have a low sensitivity to distinguish differences in physical fitness preparation (assumptions) among players in terms of field position. In the future, selected tests should take into account the higher level of specific locomotive movements (movement patterns) of each player’s function. Another option is to create new and more specific tests for individual player positions.

## Figures and Tables

**Figure 1 sports-07-00046-f001:**
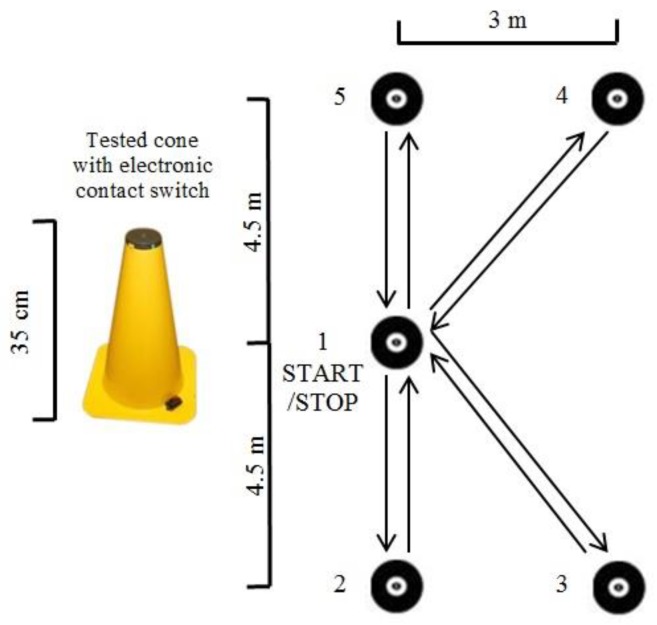
Agility K-test.

**Figure 2 sports-07-00046-f002:**
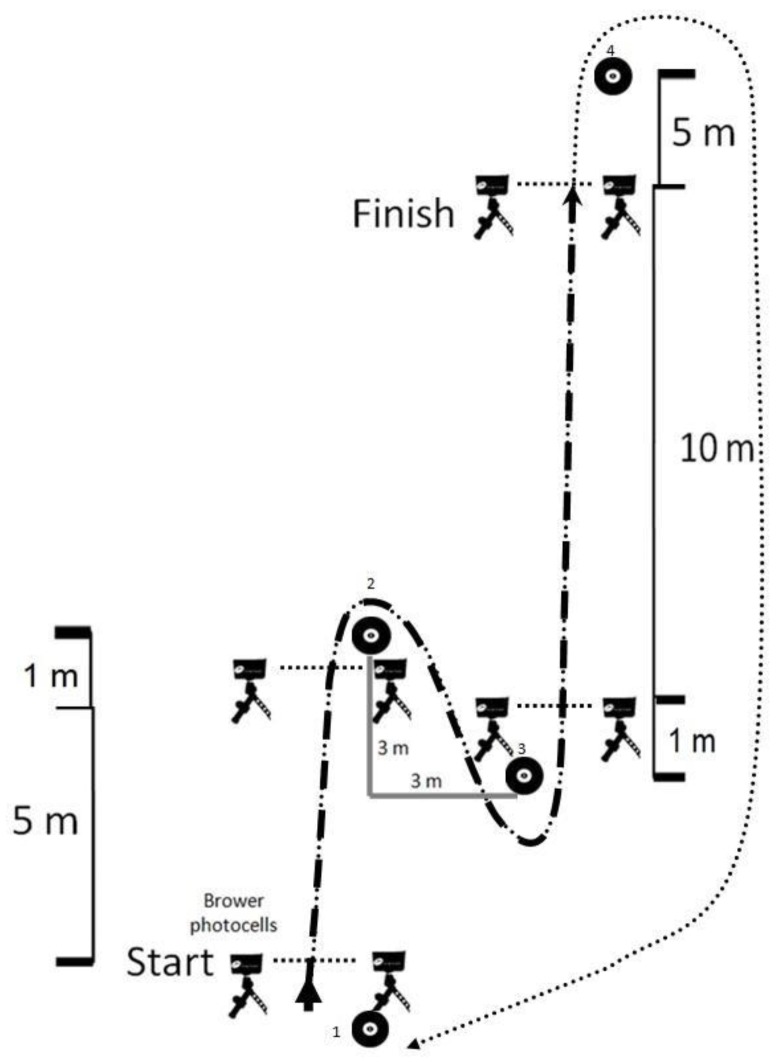
Repeated sprint ability (RSA).

**Table 1 sports-07-00046-t001:** Differences in group fitness parameters according to playing position. Data values are expressed in the following format: Mean (standard deviation).

Parameters	GK	FB	CD	WM	CM	AT	ANOVA		Post–Hoc Test
	*n* = 9	*n* = 25	*n* = 15	*n* = 27	*n* = 25	*n* = 22	F	*p*	
**BH** **(cm)**	182.44 (5.80)	178.12 (6.43)	184.98 (4.84)	177.18 (7.34)	173.54 (4.48)	179.94 (5.82)	8.28	<0.01	CD vs. FB,WM,CM CM vs. GK,CD,AT
**BW** **(kg)**	73.3 (6.68)	65.77 (6.37)	73.71 (6.19)	67.37 (10.96)	63.14 (6.13)	71.62 (6.72)	5.99	<0.01	CM vs. GK,CD,AT FB vs. CD
**S5** **(s)**	1.13 (0.07)	1.13 (0.06)	1.10 (0.05)	1.13 (0.07)	1.13 (0.08)	1.16 (0.06)	0.86	>0.05	-
**S10** **(s)**	1.90 (0.10)	1.88 (0.09)	1.85 (0.06)	1.89 (0.09)	1.90 (0.10)	1.90 (0.08)	0.73	<0.056	-
**F20** **(s)**	2.55 (0.11)	2.42 (0.10)	2.38 (0.09)	2.35 (0.08)	2.47 (0.11)	2.44 (0.13)	6.91	<0.01	GK vs. FB, CD,WM WM vs. CM, AT
**A5050D** **(s)**	2.59 (0.11)	2.53 (0.10)	2.54 (0.11)	2.57 (0.32)	2.47 (0.14)	2.48 (0.09)	1.36	>0.05	-
**A505N** **(s)**	2.61 (0.23)	2.53 (0.10)	2.53 (0.12)	2.47 (0.13)	2.44 (0.08)	2.48 (0.07)	4.02	<0.01	GK vs. WM, CM
**K-test** **(s)**	10.85 (0.40)	10.87 (0.43)	10.80 (0.30)	10.86 (0.69)	10.78 (0.49)	10.88 (0.34)	1.52	>0.05	-
**RSA** **(s)**	6.05 (0.18)	5.46 (0.19)	5.68 (0.19)	5.54 (0.15)	5.68 (0.22)	5.56 (0.20)	15.11	<0.01	GK vs. FB,CD,WM,CM,AT FB vs. CD,CM
**YYRT1** **(m)**	1815.11 (378.38)	2275.52 (205.34)	2220.27 (284.36)	2229.04 (192.32)	2232.32 (251.39)	2236.00 (192.65)	5.54	0	GK vs. FB,CD,WM,CM,AT
**VO2max (mL/min/kg)**	54.27 (5.04)	59.62 (3.91)	59.29 (3.29)	59.95 (2.87)	60.71 (3.42)	58.84 (3.67)	4.63	0	GK vs. FB,CD,WM,CM,AT

Legend: GK—goalkeeper, FB—full back, CD—center back, WM—wide midfield, CM—central midfield, F—forward, BH—body height, BW—body weight, S5—5-m sprint, S10—10-m sprint, F20—20-m sprint, A505D—agility 505 dominant, A505N—agility 505 non-dominant.

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
