# Peer review of "Physical Fitness Characteristics of High-level Youth Football Players: Influence of Playing Position"

_sports, 2019, doi:10.3390/sports7020046_

Round 1
Reviewer 1 Report
The present study aimed to investigate the differences in the various 57 physical fitness characteristics of high-level youth football players. The findings are novel and the study provides interesting data concerning the physical qualities of youth players. Moreover, the authors should be congratulated for their good work. However, some minor flaws have been detected and have to be addressed. These flaws are related to the presentation of your results . For example, you provide a Table containing all your results. I would also prefer to include a Figure (or Figures) representing your most important findings. In addition, did your ran any correlations between your variables? If so, this must be included in your results section and the associations between variables should be discussed in the respective section. Please also include a Figure (or figures) representing some of your most important associations. For instance, it is known that repeated sprint ability is significantly associated with aerobic capacity. Did you ran this correlation in your variables? A short paragraph including some practical applications derived from the study has to be included.
Abstract
line 16: change to "played"
Results
line: 143-145: Is this sentence necessary? Probably not.
Discussion
lines 186-189: Is this information necessary? Please remove the sentence.
line 192 "had an aim of": please make this sentence more clear.
Author Response
Dear Reviewer,
Thank you for the time and effort in reviewing this paper. Please see below for our response and the suggested revisions.
Thank you for all suggestions, advices and comments. Each of them are very usefull and should improve our manuscript.
Please read attached file our responses to Reviewer 1.
Thank you in advance.
Authors

Reviewer 2 Report
This study examines abilities of high-level youth football players. It should be clarified whether the participants were familiarized for the testing procedures.
L39. Change “[10, 5, 11] “ to “[5, 10, 11]”.
L46. The authors seem to suggest that the greater adaptation of midfielders is due to game-requirements. The authors should provide evidence that game performance of midfielders contributes to physical abilities. Are they not the consequence of the physical training strategies.
L50. Change “[27, 28, 11]and” to “[11, 27, 28] and “
L68. Change “bytheir” to “by their”
Ls 68-69. Please be consistent with providing positions and abbreviations.
Ls 124-126. If Bonferroni correction was used then is the P-value of <0.05 still applicable?
Ls 138-139. Were FB and CB observations for F20 performance exactly the same?
L143. Were repeated sprint ability the same for CD and CM?
Ls 144-145. Remove “Section may be divided by subheadings. It should provide a concise and precise description of the experimental results, their interpretation as well as the experimental conclusions that can be drawn”
Table 1. P-values of 0? Please provide a smaller than value.
L69 and Table. N for WD is different.
L69 Should WD be WM. Please ensure correct use of the abbreviations.
L154. Change “springs” to “sprints”
L188. Different fonts seem to be used.
Author Response
Dear Reviewer,
Thank you for the time and effort in reviewing this paper. Please see below for our response and the suggested revisions.
Thank you for all suggestions, advices and comments. Each of them are very usefull and should improve our manuscript.
Please read attached file our responses to Reviewer 2.
Thank you in advance.
Authors

Reviewer 3 Report
See the attached document.

Author Response
Dear Reviewer,
Thank you for the time and effort in reviewing this paper. Please see below for our response and the suggested revisions.
Thank you for all suggestions, advices and comments. Each of them are very usefull and should improve our manuscript.
Please read attached file our responses to Reviewer 3.
Thank you in advance.
Authors

Round 2
Reviewer 3 Report
I Would like to congratulate the authors for this revised version of the paper.